# Inter-limb asymmetry in youth elite soccer players: Effect of loading conditions

Giampiero Ascenzi[1,2]*, Cristoforo Filetti[3,4], Valter Di Salvo[1,5], F. Javier Nuñez[2], Luis Suarez-Arrones[2,6], Bruno Ruscello[3,7], Fabio Massimo Francioni[1], Alberto Mendez Villanueva[8]

1 Football Performance & Science Department, Aspire Academy, Doha, Qatar, 2 Physical Performance and Sport Research, Pablo de Olavide University, Sevilla, Spain, 3 Italy School of Sport Sciences and Exercise, Faculty of Medicine and Surgery, "Tor Vergata" University, Rome, Italy, 4 Paris Saint Germain Fc performance department, Paris, France, 5 Department of Movement, Human and Health Sciences, University of Rome "Foro Italico", Rome, Italy, 6 Performance Department, FC Basel, Basel, Switzerland, 7 School of Sports and Exercise Sciences, "San Raffaele" University, Rome, Italy, 8 Qatar Football Association, Doha, Qatar

* giampiero.ascenzi@hotmail.it

**Editor:** Riccardo Di Giminiani, University of L'Aquila Department of Clinical Sciences and Applied Biotechnology: Universita degli Studi dell'Aquila Dipartimento di Scienze Cliniche Applicate e Biotecnologiche, ITALY

## Abstract

The presence of inter-limb asymmetries can influence strength performance and represent an injury risk factor for team sport athletes. The present study aimed to investigate the effects of changes in resistance loads using different assessment modalities on the magnitude and the direction of inter-limb asymmetry within the same leg. Fifteen young elite soccer players from the same professional academy performed rear-foot-elevated-split-squat-test at different loading conditions (body mass with no overload, 25% of body mass, 50% of body mass 50%), isokinetic knee flexor (concentric 30˚·s⁻¹, concentric 60˚·s⁻¹, eccentric 90˚·s⁻¹) and extensor (concentric 60˚·s⁻¹, eccentric 60˚·s⁻¹). The outcomes from the agreement analyses suggested moderate level agreement between body mass vs body mass 25% (Kappa = 0.46), with no agreement or fair agreement for the other between-assessment comparison. Our results demonstrated that the magnitude and direction of within-limb strength imbalances were inconsistent when compared within the same assessment under different resistance load conditions.

## Introduction

Soccer performance is characterized by unforeseeable and explosive actions involving sprinting, jumping, tackling, kicking, turning, and changes of pace [1]. The multi-planar and unilateral nature of these soccer-specific actions require the production of force and power in single leg conditions [2, 3]. The involvement in this type of movement patterns renders players prone to the development of strength imbalances, with implications from clinical and applied perspectives [4, 5]. The notion of imbalance refers to inter-limb asymmetry as the difference in performance or function of one limb relative to the other [6].

A number of studies suggested that players with substantial inter-limb asymmetries are at higher risk of injury [7, 8]. In this context, Croisier et al., found players with significant

**Data Availability Statement:** All Supporting files are available from the excel database attached to the manuscript.

**Funding:** The author(s) received no specific funding for this work.

**Competing interests:** The authors have declared that no competing interests exist.

isokinetic knee flexors strength imbalances had a five-time hamstring injury risk [8]. Likewise, consideration of strength imbalances is also relevant to the return to play process [7, 9]. Despite the influence of inter-limb asymmetries on injury risk in soccer, findings on their role on performance are inconsistent [10–12]. Researchers observed no association between strength imbalances in counter movement jump height and linear sprint performance both in male and female youth professional soccer players [10, 11]. In contrast, Coratella et al., found a positive association between strength imbalances in quadriceps and hamstrings isokinetic strength at different angular velocities (30˚/s and 300˚/s) and performance on change of direction tests [12].

Despite isokinetic strength testing is a very commonly used method to identify strength deficits and associated inter-limb asymmetries in soccer players [5, 7–9, 13], it requires sophisticated procedures and requires costly and non-portable equipment, which limits its use in the daily practice of most soccer teams. Thus, in addition to isokinetic screening, other field-based strength testing protocols are widely implemented in daily practice for assessing unilateral muscular strength, and associated inter-limb asymmetries, to aid in guiding training, rehabilitation and injury prevention. For instance, the rear-foot-elevated-split-squat-test (RFESS) can be regarded as a practical method for measuring unilateral strength in sport [14–17].

While the notion of magnitude remains important, determining the direction of strength imbalances is also relevant to understanding the consistency of the regarding the preference of one side compared the other one [18, 19]. In this regard, inconsistencies in the direction of inter-limb asymmetries with different strength assessments have been previously reported [18, 19]. Recently, Bishop et al. highlighted how the levels of agreement on the same side were low during different isometric and ballistic strength assessments in recreational sports athletes, emphasizing the need for an individualised approach to quantify an asymmetry [20, 21]. Despite the increasing use of different tests to monitor inter-limb asymmetry magnitude and direction, the relationship and interchangeability between their outcomes is not clear. Thus, the present investigation aimed to evaluate the relationship of the magnitude and the direction of inter-limb asymmetries measured with different strength testing modes and/loads in a sample of elite youth soccer players.

## Materials and methods

Players were assessed using RFESS, isokinetic knee flexor (IKF) and isokinetic knee flexor (IKE) protocols executed in different loading conditions. Technical coaches gathered information regarding a player's limb dominance prior to assessment [20]. The inter-limb asymmetries for both tests were determined as [22]:

$$Asymmetry\ (\%) = 1 - [(non - dominant\ leg/dominant\ leg) * 100]$$

Test overload was selected according to the subject body mass for the RFESS assessment, whereas use of the lean muscle mass informed isokinetic assessments [23].

### Participants

Fifteen, full-time, elite youth male soccer players from Qatar National Team (Doha, Qatar) took part in the study (age: 18.5 ± 0.6 years; height: 174.7 ± 6.3 cm; body weight 66.8 ± 6.8 Kg). Players had ~8 training sessions per week involving strength, aerobic fitness and soccer, with one national league and two international friendlies scheduled weekly and every four weeks during the study period. Of the available sample, five players did not fulfil the eligibility criteria and were excluded due to previous knee or chronic lower limb injures. Signed consent was

obtained use data for research purposes, with this study data collection as part of a project was approved by the Qatar Antidoping Lab (E2013000004).

## Experimental design

Dual Energy X-Ray Absorptiometry (DEXA) assessed lean limb mass of study participants (Lunar iDXA enCORE 2008 GE Medical Systems Lunar Version12.30.008, 3030 Ohmedo Drive, Madison, WI 53718, USA). Strength assessments were administered on two occasions 72-hour apart as a washout period. Once the body mass of each player was detected (SECA, Hamburg, Germany), the individual values at 25% (BM25%) and 50% (BM50%) of body mass were calculated to define the appropriate external loads to be applied in the RFESS. The extra weight added to reach BM25% and BM50% including the weight of the lifting bar (7.3 Kg). On day 1, after a standardized warm up (5-min cycling, 3 -min ballistic stretching, 1 x 3 trials at BM25% & BM 50% back squat in unilateral condition), subjects performed 3 maximal, unilateral repetitions assessed using the RFESS on the Smith Machine (Multipower, Technogym™, Gambettola, Italy) lifting exclusively the bar (Fig 1). The same test was repeated 5 and 10 minutes after with a total extra load of BM25% and BM50% (Fig 1). A professional goniometer (Baseline® Measurement, White Plains, NY, USA) was used to assess the 90˚ knee angle which started the maximal ballistic push off (positive phase, upward extension) [24]. The best mean power trials for each load were recorded [24, 25]. All assessments were recorded using a linear encoder (SmartCoach™, EuropeAB, Stockholm, Sweden).

On day 2, after a10 min of warm-up cycling at 100 W, the isokinetic peak torque of the knee flexor and extensor muscles was assessed by an isokinetic dynamometer (Fig 2) (CSMi, Stoughton, MA) [26]. All the participants performed three unilateral trials with dominant leg

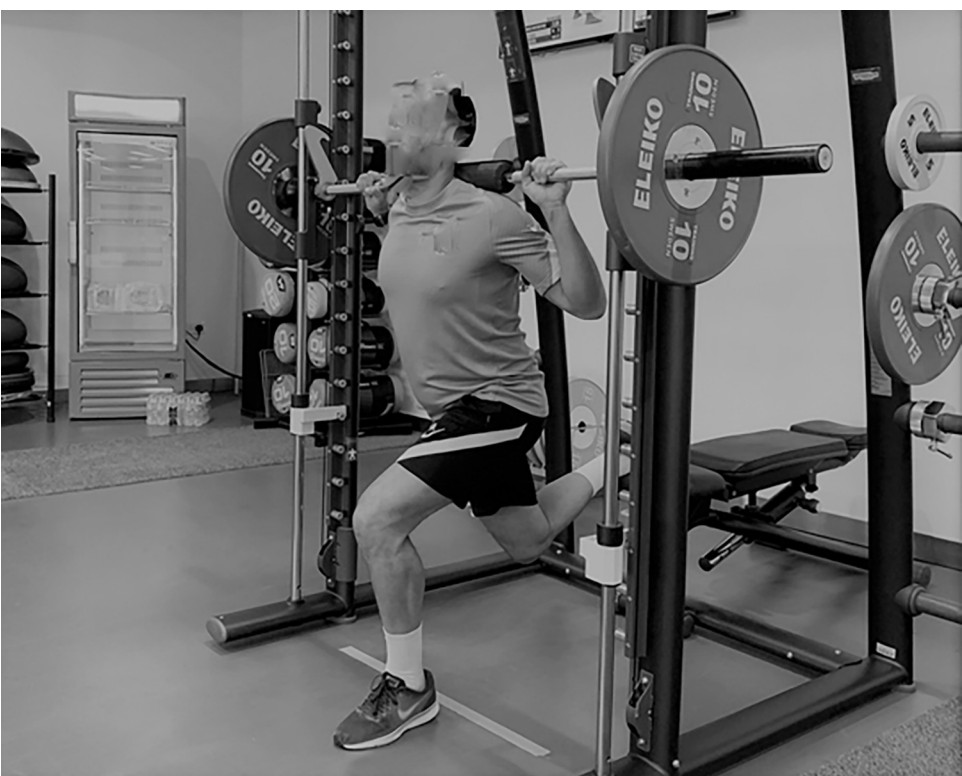

**Fig 1. Rear Foot Elevated Split Squat.**

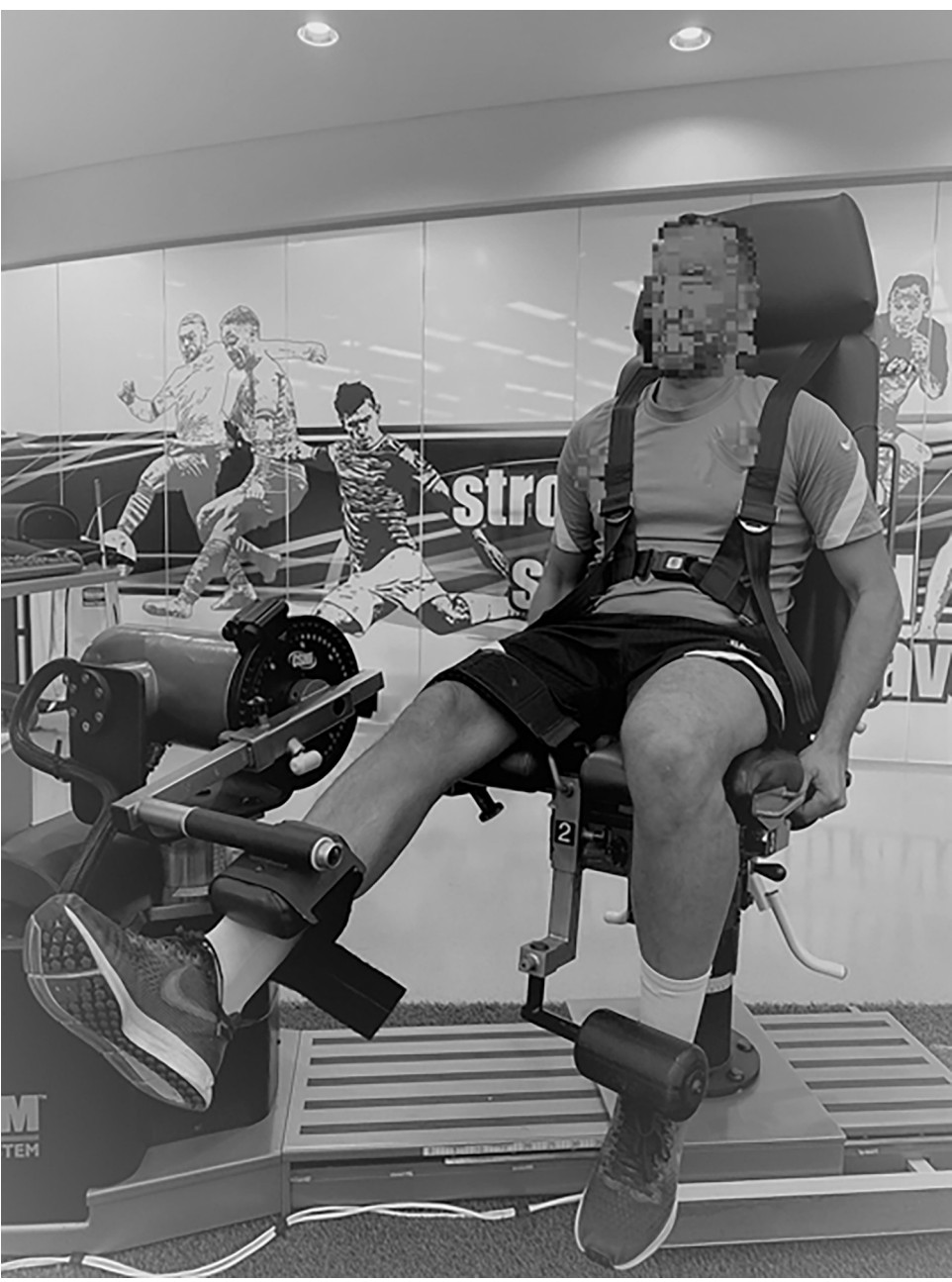

**Fig 2. Isokinetic knee flexion and knee extension.**

(DL) and non-dominant leg (NDL). The peak torque was observed for knee flexor in concentric actions at $30°·s^{-1}$ and $60°·s^{1}$ angular velocities, and at $90°·s^{-1}$ angular velocities for the eccentric actions. Peak torque of knee extension during concentric actions at $60°·s^{-1}$ angular velocities and eccentric $60°·s^{-1}$ angular velocities have been analyzed. The range of movement was recorded from $90°$ to $0°$ and from $0°$ to $90°$, with 0 defining full knee extension [27], with the hip flexion angle fixed at $90°$. The best peak torque trials (N·m) for each angular velocity, was recorded and subsequently normalized for the lean muscle mass of the considered lower limb (N·m·Kg$^{-1}$) [28].

## Statistical analysis

Visual inspection of frequency distributions of the raw strength assessment data suggested assumptions of normality were not violated. Data are presented as mean ± standard deviation (SD), plus minimum and maximum for descriptive and strength-related variables. The isokinetic peak torque values were normalized ($N·m·Kg^{-1}$) for each trial. The intraclass correlation coefficients (ICCs) determined relative and absolute reliability of the measurements. T-test was used to determine the difference between DL and NLD during RFESS, IKF and IKE. Kappa coefficient was calculated to determine the levels of agreement between asymmetries for a common metric across the same tests in different conditions tests and values were interpreted as follow, 0.01–0.20 = slight; 0.21–0.40 = fair; 0.41–0.60 = moderate; 0.61–0.80 = substantial; 0.81–0.99 = nearly perfect [29]. A pre-defined threshold ≥15% determined a substantial asymmetry between limbs [30]. Scatter plots illustrated the relationship between DL versus NDL asymmetries over the average of the two measurements. Statistical significance was set a priori at $p < 0.05$. All analyses were conducted using IBM SPSS 25.0.

## Results

Descriptive data for the different strength assessments are displayer in Table 1. IKF concentric $30°·s^{-1}$ showed moderate reliability (ICC = 0.67). The outcomes for agreement analyses are presented in Table 2, with moderate level agreement between BM vs BM25% (Kappa = 0.46). In the REFFS assessment, players did not show substantial asymmetries irrespective of the condition (Fig 3). Likewise, the direction and magnitude of strength imbalances between limbs was inconsistent in both IKF and IKE assessments (Fig 4).

## Discussion

This is the first study to explore within-assessment mode inter-limb asymmetries under different loading conditions and the relation between inter-limb. Our main findings revealed that

**Table 1. Descripitve information for dominant and non-dominant leg average power and peak torque, level of asymmetry (%) and reliability in Rear Foot Elevated Split Squat (RFESS), Isokinetic Flexor and Isokinetic Extensor.**

|  | Dominant Leg | Non-Dominant Leg | P Value | Asymmetry (%) | ICC (95%CI) |
|---|---|---|---|---|---|
| **RFESS** |  |  |  |  |  |
| *AVG Power [W]* |  |  |  |  |  |
| BM | 682.5 ± 77.9 | 714.7 ± 78.6 | 0.02 | 6.6 ± 3.7 | 0.89 (0.69–0.97) |
| BM25% | 737.1 ± 85.1 | 744.1 ± 84.8 | 0.59 | 4.8 ± 3.1 | 0.92 (0.76–0.97) |
| BM50% | 731.4 ± 88.2 | 729.3 ± 89.7 | 0.33 | 5.2 ± 3 | 0.91 (0.72–0.97) |
| ***Isokinetic Flexor*** |  |  |  |  |  |
| *Peak Torque [N·m·Kg$^{-1}$]* |  |  |  |  |  |
| Concentric 30˚/s | 131.5 ± 20.4 | 125.4 ± 18.2 | 0.32 | 12.8 ± 7.3 | 0.67 (0.02–0.89) |
| Concentric 60˚/s | 130.3 ± 19.6 | 126.9 ± 17.2 | 0.05 | 4.5 ± 4.3 | 0.97 (0.91–0.99) |
| Eccentric 90˚/s | 150.4 ± 29.5 | 145.4 ± 29.5 | 0.37 | 7.1 ± 13.2 | 0.83 (0.51–0.94) |
| ***Isokinetic Extensor*** |  |  |  |  |  |
| *Peak Torque [N·m·Kg$^{-1}$]* |  |  |  |  |  |
| Concentric 60˚/s | 190.7 ± 27 | 196.6 ± 32.3 | 0.38 | 6.7 ± 5.6 | 0.89 (0.69–0.96) |
| Eccentric 60˚/s | 251.6 ± 62.5 | 238.1 ± 72.7 | 0.31 | 11.8 ± 10.3 | 0.94 (0.51–0.94) |

BM = RFESS = Rear Foot Elevated Split Squat, Body mass, BM25% = Body mass 25%, BM50% = Body Mass 50%, AVG Power = Average power, ICC = Intraclass Correlation Coefficient.

Table 2. Kappa coefficients and descriptive levels of agreement showing how consistently inter-limb asymmetry favors the same limb within the same assessment.

| | | Kappa Coefficient | Level of Agreement |
|---|---|---|---|
| **RFESS** | | | |
| *AVG Power [W]* | | | |
| | BM vs BM25% | 0.42 | Moderate |
| | BM25% vs BM50% | 0.16 | Slight |
| | BM vs BM50% | -0.72 | No Agreement |
| **Isokinetic Knee Flexion** | | | |
| *Peak Torque [N·m·Kg$^{-1}$]* | | | |
| | Concentric 30˚/s vs Concentric 60˚/s | -0.75 | No Agreement |
| | Concentric 30˚/s vs Eccentric 90˚/s | 0.12 | Fair |
| | Concentric 60˚/s vs Eccentric 90˚/s | 0.12 | Fair |
| **Isokinetic Knee Extension** | | | |
| *Peak Torque [N·m·Kg$^{-1}$]* | | | |
| | Concentric 90˚/s vs Eccentric 90˚/s | -0.75 | No Agreement |

RFESS = Rear Foot Elevated Split Squat, BM = Body Mass, BM25% = Body Mass 25%, BM50% = Body Mass 50%, AVG Power = Average Power.

the magnitude and direction of inter-limb asymmetries were inconsistent in our sample of youth soccer players, and the detection of asymmetries presented large variations in the same player depending on the load imposed on the strength test employed. Importantly, studies in this field highlighted that the extent of any potential inter-limb asymmetry can be prone to the protocol selection given the number of equations proposed in the sports performance literature [20]. In our investigation, we used the formula by Schiltz and colleagues [22] since we elected to assume a practical differentiation between the dominant and non-dominant leg given the unilateral and multiplanar nature of soccer performance [2, 3]. Furthermore, the notion of direction interpreted and quantified as any asymmetry favouring the same limb between metrics and/or tasks is another important outcome we described in our report informed by the current knowledge base in this field [18, 19].

With the majority of the players we screened showing heterogeneity of patterns in each loading condition (Figs 1 and 2), performing assessment under different loading schemes (i.e., speed and/or external load) and turned out to be a useful and alternative approach for quantifying asymmetries. This is critical considering that strength asymmetries have been identified as potential performance impairment factor [6, 12] as well as a risk factor for different muscle and joint injuries [8, 31]. Moreover, strength asymmetries are often assessed to guide the rehabilitation and return to play process in different kind of soccer-related injuries 5/12/21 5:24:00 PMk. While the consistency of the magnitude and direction of strength asymmetries have been previously investigated [18], this study adds to the existing knowledge by showing for the first time at individual level a lack of agreement between the asymmetries obtained using the same test but with different external loads. Due to the typical poor agreement observed between the different loads employed here to assess strength asymmetries while using the same test, it appears challenging to use strength measurements derived from a single test and load as a surrogate of overall strength asymmetries in that given assessment.

The limited agreement between the different isokinetic hamstring strength asymmetries is another important finding of the present study. Since hamstring strength deficits have been identified as both risk factors and post-injury consequence [8, 32], the data presented in this study might have implications for coaches and therapists dealing with both injured and non-injured players. Our findings suggest that a strength asymmetry detected using a single load and/or contraction mode in a given test should not be regarded a proxy for inter-limb

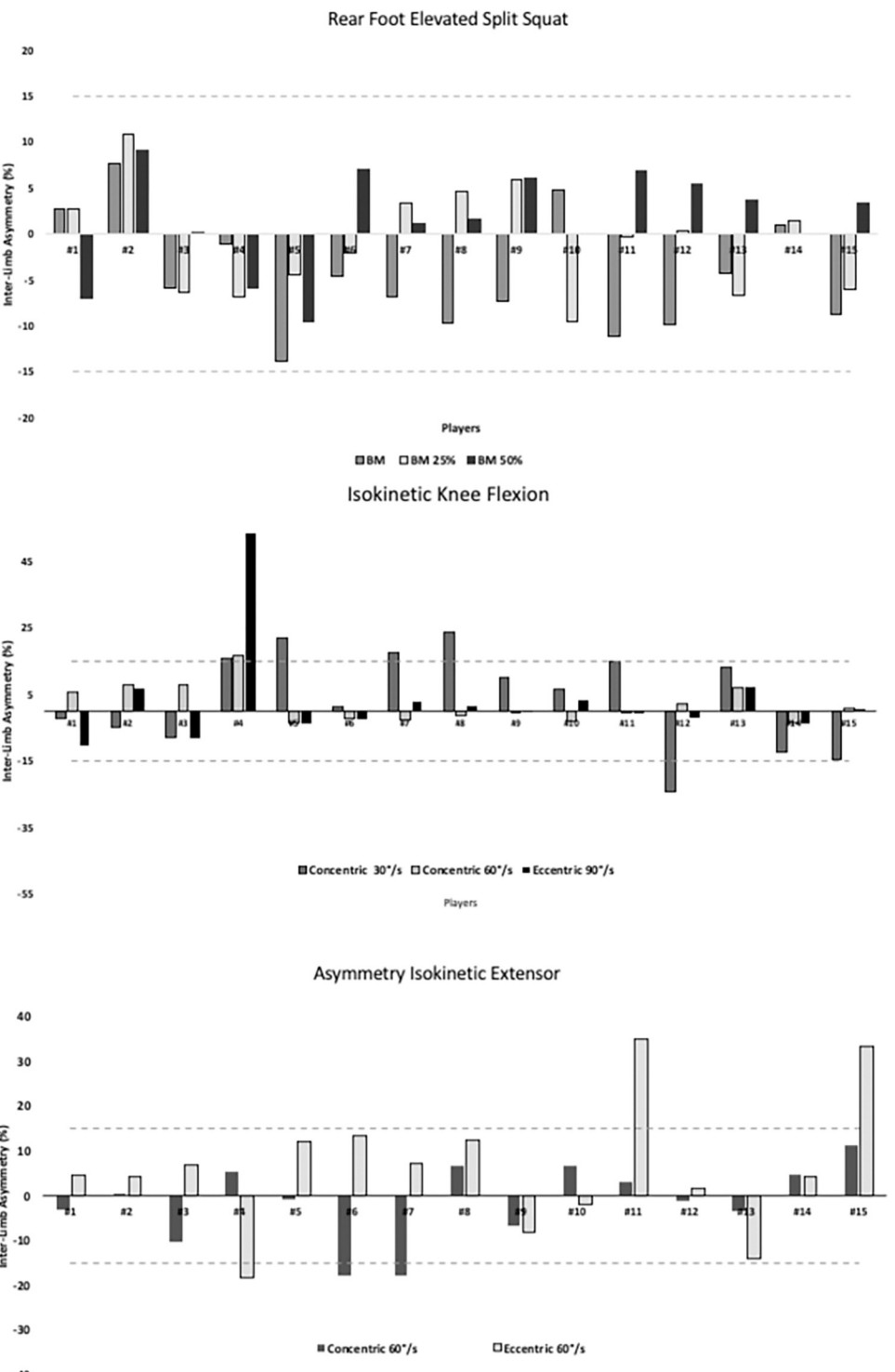

**Fig 3. Magnitude and direction of inter-limb asymmetry in Rear Foot Elevated Split Squat, isokinetic flexion and isokinetic extension assessments.** BM = Body mass, BM25% = Body mass 25%, BM50% = Body Mass 50%. *Note*: above 0 indicates asymmetry favours the dominant leg and below 0 indicates asymmetry favours the non-dominant leg.

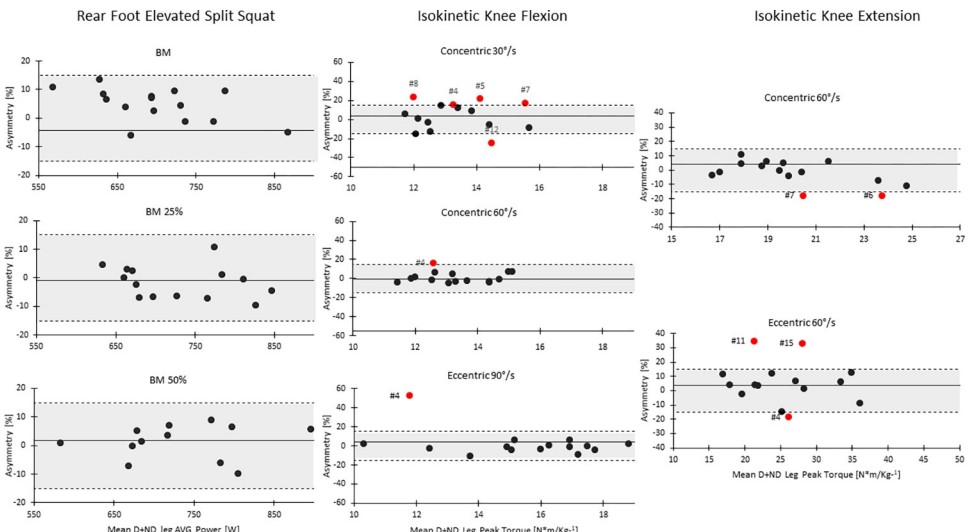

**Fig 4. Mean power, mean force, magnitude and direction of inter-limb asymmetry in Rear Foot Elevated Split Squat, isokinetic flexion and isokinetic extension assessments.** Empty circles represent players with asymmetry > 15%. BM = Body mass, BM25% = Body mass 25%, BM50% = Body Mass 50%, DL = Dominant Leg, NDL = Non-dominant leg, AVG Power = Average power.

hamstring asymmetry. Similarly, strength asymmetries and imbalances detected using isokinetic knee extension and RFESS have been suggested as prognostic factors relevant to physical performance, injury prevention, injury rehabilitation and re-injury risk following the return to play process [9, 12, 16, 33, 34]. Thus, according to present findings performance and clinical practitioners should be aware that unilateral strength assessment and associated asymmetries and imbalances appear too complex in nature to be amenable to a single diagnostic assessment [34]. In line with previous investigations [18, 19, 21], our results confirmed how quantifying the presence any potential inter-limb asymmetry requires consideration of different test protocols within a multi-assessment framework relevant to the context under examination. Specifically, the scrutiny of a single strength assessment in isolation is unlikely to result in understanding the true direction and magnitude of any potential inter-limb asymmetry. Consequently, single training approaches to reduce strength asymmetries, as performance enhancement, injury prevention and rehabilitation are unlike to be successful in the long term. Practically, in the worst case, the effect of an intervention could be considered either insignificant or highly appropriate for strength asymmetries correction depending on specific outcomes derived from the chosen test and loading condition.

Despite the context of our study, the small size of the sample of player we examined might be considered as the main limitation of our study. Furthermore, the lack of previous research adopting the same design and procedures, particularly in the selection of loads, limits comparisons with players from other populations. Also, the practical barriers of performing 50%BM assessment for some players in our sample may be relevant to design of future research in this field regarding consideration of this particular assessment condition.

## Conclusions

No previous research examined the magnitude and the direction of the inter-limb asymmetry using the same test with different loads. The degree of within-limb strength imbalances under different loading conditions may depend on the assessment mode chosen. The a priori adoption of a single assessment is unlikely to identify the presence of an inter-limb asymmetry

being generalizable from a practical standpoint. Accordingly, consideration of different assessment methods relevant to the determination of a potential inter-limb asymmetry is required to obtain information relevant to the development strength and conditioning programs for the individual player.

## Supporting information

**S1 File. Assessments database.**
(XLSX)

## Acknowledgments

We wish to thank the players and coaches for their participation in this study.

## Author Contributions

**Conceptualization:** Giampiero Ascenzi, Fabio Massimo Francioni, Alberto Mendez Villanueva.

**Data curation:** Giampiero Ascenzi, Fabio Massimo Francioni.

**Formal analysis:** Giampiero Ascenzi.

**Funding acquisition:** Giampiero Ascenzi.

**Investigation:** Giampiero Ascenzi, Alberto Mendez Villanueva.

**Methodology:** Giampiero Ascenzi, Alberto Mendez Villanueva.

**Project administration:** Giampiero Ascenzi.

**Resources:** Giampiero Ascenzi, Cristoforo Filetti.

**Supervision:** Alberto Mendez Villanueva.

**Writing – original draft:** Giampiero Ascenzi, Fabio Massimo Francioni.

**Writing – review & editing:** Valter Di Salvo, F. Javier Nuñez, Luis Suarez-Arrones, Bruno Ruscello, Fabio Massimo Francioni, Alberto Mendez Villanueva.

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
