## [Decision Letter · Decision Letter 0]

2 Feb 2022

PONE-D-21-30024Inter-limb asymmetry in youth elite soccer players: effect of loading conditionsPLOS ONE

Dear Dr. Ascenzi,

Thank you for submitting your manuscript to PLOS ONE. After careful consideration, we feel that it has merit but does not fully meet PLOS ONE’s publication criteria as it currently stands. Therefore, we invite you to submit a revised version of the manuscript that addresses the points raised during the review process.

We look forward to receiving your revised manuscript.

Kind regards,

Riccardo Di Giminiani

Academic Editor

PLOS ONE

Journal Requirements:

3. Thank you for including your ethics statement:  "The study was approved by the local institutional review boarder.".   

6. We note that Figure 1 and 2 includes an image of a participant. 

Reviewers' comments:

Reviewer's Responses to Questions

**Comments to the Author**

1. Is the manuscript technically sound, and do the data support the conclusions?

Reviewer #1: Yes

Reviewer #2: Yes

2. Has the statistical analysis been performed appropriately and rigorously? 

Reviewer #1: Yes

Reviewer #2: No

3. Have the authors made all data underlying the findings in their manuscript fully available?

Reviewer #1: Yes

Reviewer #2: Yes

4. Is the manuscript presented in an intelligible fashion and written in standard English?

Reviewer #1: Yes

Reviewer #2: No

5. Review Comments to the Author

Reviewer #1: Abstract

“rear-foot-elevated-split-squat-test (body mass, body mass 25% body mass 50%)”. Reading the following part of the article, “body mass 25%” mean “25% of the body mass”. If this interpretation is correct, it could be clearer reported.

Participants

Some information about the nationality of the observed players could be reported to better define the experimental sample, because players’ competitive level could also depend by the national context.

Results

Table 1 and 2 should be better structured and readable, by reporting rows which are able to define the subareas of results.

S3 and S4 Figs could be better visible (higher definition)

Discussion

Line 193. Regardless of the number of participants, repeated measures on the same trials, along different phase of a season, and controlling the players internal training load, could enlarge the point of view on the magnitude and the direction of the inter-limb asymmetry using the same test with different loads.

Reviewer #2: L44: Full stop mark is missing just after the reference.

L55-56: This section requires a more in-depth explanation. Why rear-foot-elevated-split-squat-test (RFESS) can be regarded as a practical method for measuring unilateral strength in sport?

L69: "IKF" and "IKE" acronyms are reported here for the first time. Please explain the meaning within the text.

L92-93: What was the range of movement of knee flexion and how did the authors assessed it during the RFESS? Please report this information.

L98: Warm up is the same on Day 2 as on Day 1? Please report this information.

L100: "DL" and "NDL" acronyms are reported here for the first time. Please explain the meaning within the text.

L100-103: Please report the references to support the choice of the angular velocities that have been used for the isokinetic test.

L103-104: Please report the hip angle or if in an upright position, the backrest angle.

L115-116: Did the authors assumed normal distribution or did they verify it in order to chose the most suitable statistical hypothesis test? Please report this information.

L136: Data in Table 2 look to be inconsistent. Values of Dominant and Non-dominant leg are reported in column 2 and 3 but at L72 the asymmetry was calculated between stronger and weaker limb. Is always dominant leg the stronger and the non-dominant the weaker?

L163-164: Reference is required for this statement.

L151: Discussion section should be largely improved. For instance, the aim of this study is "...to evaluate the relationship of the magnitude and the direction of inter-limb asymmetries measured with different strength testing modes and/loads in a sample of elite youth soccer players". The 1) magnitude is evaluated by the inter-limb asymmetries formula at L72 but it is not clear how do you evaluate 2) the "direction of the inter-limb asymmetries" and 3) the relation between 1) and 2). Please clarify this point.

L174-179: This paragraph does not make much sense and it should be rewritten. It seems more review of the literature about the available methods to assess strength asymmetries and imbalances rather than fully expressing the “whys” of this study. Authors should also focus on better explaining their results by discussing the task-specificity of inter-limb asymmetries and the need that more than a single test should be used to profile muscular imbalances (e.g., Bishop C, Lake J, Loturco I, Papadopoulos K, Turner A, Read P. Interlimb asymmetries: The need for an individual approach to data analysis. J Strength Cond Res, 2018; (May): 1–7)

L179, 182: Full stop mark is missing just after the reference.

L188: This sentence should be rephrased by avoiding negations (e.g., Nevertheless, limitations in our study are...)

6. PLOS authors have the option to publish the peer review history of their article (what does this mean?). If published, this will include your full peer review and any attached files.

Reviewer #1: **Yes: **Corrado Lupo

Reviewer #2: **Yes: **Alexandru Nicolae Ungureanu

---

## [Author Response · Author response to Decision Letter 0]

6 Apr 2022

I would like to say thanks for the points that the reviewers highlighted on our manuscript. 

We have attended to all of the reviewers suggestions.

---

## [Decision Letter · Decision Letter 1]

22 Apr 2022

PONE-D-21-30024R1Inter-limb asymmetry in youth elite soccer players: effect of loading conditionsPLOS ONE

Dear Dr. Ascenzi,

Thank you for submitting your manuscript to PLOS ONE. After careful consideration, we feel that it has merit but does not fully meet PLOS ONE’s publication criteria as it currently stands. Therefore, we invite you to submit a revised version of the manuscript that addresses the points raised during the review process.

We look forward to receiving your revised manuscript.

Kind regards,

Riccardo Di Giminiani

Academic Editor

PLOS ONE

Journal Requirements:

Reviewers' comments:

Reviewer's Responses to Questions

**Comments to the Author**

1. If the authors have adequately addressed your comments raised in a previous round of review and you feel that this manuscript is now acceptable for publication, you may indicate that here to bypass the “Comments to the Author” section, enter your conflict of interest statement in the “Confidential to Editor” section, and submit your "Accept" recommendation.

Reviewer #1: All comments have been addressed

Reviewer #2: All comments have been addressed

2. Is the manuscript technically sound, and do the data support the conclusions?

Reviewer #1: Yes

Reviewer #2: Yes

3. Has the statistical analysis been performed appropriately and rigorously? 

Reviewer #1: Yes

Reviewer #2: Yes

4. Have the authors made all data underlying the findings in their manuscript fully available?

Reviewer #1: Yes

Reviewer #2: Yes

5. Is the manuscript presented in an intelligible fashion and written in standard English?

Reviewer #1: Yes

Reviewer #2: Yes

6. Review Comments to the Author

Reviewer #1: (No Response)

Reviewer #2: Dear authors,

Your latest review has improved the quality of the manuscript. Please check for consistency when you address to "Non-dominant" within the text. @L120 and Tab1 you report "No-dominant" instead of "Non-dominant". Please check.

7. PLOS authors have the option to publish the peer review history of their article (what does this mean?). If published, this will include your full peer review and any attached files.

Reviewer #1: **Yes: **Corrado Lupo

Reviewer #2: **Yes: **Alexandru Nicolae Ungureanu

---

## [Author Response · Author response to Decision Letter 1]

3 May 2022

Review Comments to the Author

Please use the space provided to explain your answers to the questions above. You may also include additional comments for the author, including concerns about dual publication, research ethics, or publication ethics. (Please upload your review as an attachment if it exceeds 20,000 characters).

Reviewer #1: 

(No Response)

Reviewer #2: 

Dear authors,

Your latest review has improved the quality of the manuscript. Please check for consistency when you address to "Non-dominant" within the text. @L120 and Tab1 you report "No-dominant" instead of "Non-dominant". Please check.

Thank you for this comment. We have attended to this suggestion concerning the use of abbreviations in our manuscript.

---

## [Decision Letter · Decision Letter 2]

26 May 2022

Inter-limb asymmetry in youth elite soccer players: effect of loading conditions

PONE-D-21-30024R2

Dear Dr. Ascenzi,

We’re pleased to inform you that your manuscript has been judged scientifically suitable for publication and will be formally accepted for publication once it meets all outstanding technical requirements.

Kind regards,

Riccardo Di Giminiani

Academic Editor

PLOS ONE

Additional Editor Comments (optional):

Reviewers' comments:

Reviewer's Responses to Questions

**Comments to the Author**

1. If the authors have adequately addressed your comments raised in a previous round of review and you feel that this manuscript is now acceptable for publication, you may indicate that here to bypass the “Comments to the Author” section, enter your conflict of interest statement in the “Confidential to Editor” section, and submit your "Accept" recommendation.

Reviewer #1: (No Response)

Reviewer #2: All comments have been addressed

2. Is the manuscript technically sound, and do the data support the conclusions?

Reviewer #1: (No Response)

Reviewer #2: Yes

3. Has the statistical analysis been performed appropriately and rigorously? 

Reviewer #1: (No Response)

Reviewer #2: Yes

4. Have the authors made all data underlying the findings in their manuscript fully available?

Reviewer #1: (No Response)

Reviewer #2: Yes

5. Is the manuscript presented in an intelligible fashion and written in standard English?

Reviewer #1: (No Response)

Reviewer #2: Yes

6. Review Comments to the Author

Reviewer #1: (No Response)

Reviewer #2: (No Response)

7. PLOS authors have the option to publish the peer review history of their article (what does this mean?). If published, this will include your full peer review and any attached files.

Reviewer #1: No

Reviewer #2: **Yes: **Alexandru Nicolae Ungureanu

---

## [Editor Report · Acceptance letter]

3 Jun 2022

PONE-D-21-30024R2 

Inter-limb asymmetry in youth elite soccer players: effect of loading conditions 

Dear Dr. Ascenzi:

I'm pleased to inform you that your manuscript has been deemed suitable for publication in PLOS ONE. Congratulations! Your manuscript is now with our production department. 

Kind regards, 

on behalf of

Prof Riccardo Di Giminiani 

Academic Editor

PLOS ONE